# Comparative Pharmacokinetics and Tissue Concentrations of Flunixin Meglumine and Meloxicam in Tilapia (*Oreochromis* spp.)

Miriam Martin [1], Stephen Smith [2], Michael Kleinhenz [3], Geraldine Magnin [1], Zhoumeng Lin [4], David Kuhn [5], Shawnee Montgomery [1] and Johann Coetzee [1,*]

1 Department of Anatomy and Physiology, Kansas State University College of Veterinary Medicine, Manhattan, KS 66506, USA; miriammartin@vet.k-state.edu (M.M.); gmagnin@vet.k-state.edu (G.M.); smontgomery@vet.k-state.edu (S.M.)
2 Department of Biomedical Sciences and Pathobiology, Virginia-Maryland College of Veterinary Medicine, Virginia Tech, Blacksburg, VA 24061, USA; stsmith7@vt.edu
3 Department of Clinical Sciences, Kansas State University College of Veterinary Medicine, Manhattan, KS 66506, USA; mkleinhe@vet.k-state.edu
4 Department of Environmental and Global Health, College of Public Health and Health Professions, University of Florida, Gainesville, FL 32610, USA; linzhoumeng@phhp.ufl.edu
5 Department of Food Science and Technology, Virginia Tech, Blacksburg, VA 24061, USA; davekuhn@vt.edu
* Correspondence: jcoetzee@vet.k-state.edu; Tel.: +1-785-532-4513

**Abstract:** Evidence of pain perception in fish is well established, but analgesic use in aquaculture is limited. The objective was to investigate the comparative pharmacokinetics of flunixin administered intramuscularly (IM) and meloxicam administered IM or orally (PO) in tilapia. Two hundred and seventy fish were assigned to 1 of 3 treatment groups: flunixin meglumine IM (2.2 mg/kg); meloxicam IM (1 mg/kg); or meloxicam PO (1 mg/kg). Blood and tissue samples were collected from 6 fish per treatment at 14 time points for 10 days. Drug concentrations were determined using ultra-high-pressure liquid chromatography coupled with mass spectroscopy. Plasma concentration versus time data were analyzed with a non-compartmental approach using a commercially available software. Flunixin reached a mean maximum concentration ($C_{max}$) of 4826.7 ng/mL at 0.5 h, had a terminal half-life ($T^1/_2$) of 7.34 h, and an area under the concentration–time curve extrapolated to infinity ($AUC_{INF\_obs}$) of 25,261.62 h·ng/mL. Meloxicam IM had a $T^1/_2$ of 9.4 h after reaching a $C_{max}$ of 11.3 ng/mL at 2 h, with an $AUC_{INF\_obs}$ of 150.31 h·ng/mL. Meloxicam PO had a $T^1/_2$ of 1.9 h after reaching a $C_{max}$ of 72.2 ng/mL at 2 h, with an $AUC_{INF\_obs}$ of 400.83 h·ng/mL. Tissue concentrations of both drugs were undetectable by 9 h. Flunixin reached a sufficient plasma concentration to potentially have an analgesic effect, while meloxicam, when administered at the given dosage, likely would not.

**Keywords:** pharmacokinetics; pain; fish; analgesia

## 1. Introduction

Aquaculture is one of the largest growing sectors of the world food supply. Between 1961 and 2016, the average annual increase in global food fish consumption (3.2%) outpaced population growth (1.6%) and exceeded that of meat from all terrestrial animals combined (2.8%) [1]. In 2013, global aquaculture production accounted for approximately 50 million of the 125 million metric tons of seafood produced annually for human consumption [2]. Ornamental fish make up another large sector, with approximately 27 million ornamental fish being traded each year [3]. With the continued growth of aquaculture as well as the ornamental fish trade, and increasing scientific discussion over the potential for pain and suffering, research into the welfare of fish is vital [4].

Evidence of pain perception in fish is supported by a similar sensory system, evidence of adverse behavioral and physiological responses and normal behavior being suspended during a potentially painful event [5]. Administration of morphine to fish has been shown to significantly reduce pain-related behaviors and opercular beat rate, showing evidence that morphine can act as an analgesic in fish [6]. The Farm Animal Welfare Committee base their guidelines for farmed fish welfare on the "Five Freedoms"; therefore, the use of analgesia in aquaculture could potentially play a role in allowing for freedom from discomfort, pain, injury, and distress [7]. Analgesic use in aquaculture is limited with little information available on the properties of analgesic drugs in most fish species. Interspecies variation has been reported and the current lack of a validated approach to assessing pain in fish limits our ability to evaluate analgesic efficacy [8].

Fish pharmacokinetic studies have primarily focused on drugs used to treat infectious disease, with little attention given to analgesic drugs. Studies published on analgesic drugs have focused on clinical efficacy and pain control with less focus on drug pharmacokinetics. The analgesic class of NSAIDs inhibit the cyclo-oxygenase enzymes, reduce inflammation that accompanies tissue injury, and decrease prostaglandin production which attenuates the response of the peripheral and central components of the nervous system to noxious stimuli, which results in a reduction in the response to pain [9]. Few NSAIDs have been evaluated with respect to analgesia in fish [10]. Flunixin has been found to be an ineffective water treatment but has been determined to be effective when injected intra-peritoneally at a dose of 2.2 mg/kg in channel catfish [11]. Meloxicam, when administered intramuscularly or intravenously at a dose of 1 mg/kg, was rapidly eliminated, suggesting that clinically relevant concentrations may be difficult to maintain in Nile tilapia [12]. Neither study examined tissue residue depletion of flunixin or meloxicam in fish.

Off-label usage of dosing regimens is often extrapolated from other species; and in fish, extrapolation is not recommended because of excessive pharmacokinetic variability between species, route of administration, and drug formulation [13]. Aquaculture pharmacokinetics are also uniquely affected by environmental factors such as temperature, pH, and water salinity in which the fish are raised. Thus, the purpose of this study was to investigate the comparative pharmacokinetics of flunixin administered intramuscularly and meloxicam administered intramuscularly or orally in tilapia under a controlled environmental setting.

## 2. Materials and Methods

### 2.1. Animals

Three hundred juvenile tilapia (*Oreochromis* spp.) were obtained from a commercial aquaculture facility (Blue Ridge Aquaculture, Martinsville, VA, USA) and maintained at the Aquaculture Lab of the Department of Food Science and Technology at Virginia Tech. The pharmacokinetic trial was performed using an indoor freshwater RAS equipped with twelve one-meter diameter polyethylene tanks (~340 L each), bubble bead filter for mechanical-solids filtration, fluidized-bed bioreactors for biological treatment to nitrify ammonia and nitrite to nitrate, UV disinfection units, heating element, and distributed diffuse aeration. Sodium bicarbonate was supplemented to the system to maintain sufficient alkalinity levels for efficient nitrification.

Water quality was rigorously monitored during the fish trial. All water quality parameters were analyzed using methods approved and adapted from [14]. Dissolved oxygen and pH were measured using a YSI ProODO (Yellow Spring Instruments, Yellow Springs, OH, USA) and an Accumet AB15 (Accumet Instruments Pte Ltd., Singapore) meters, respectively. Temperature was monitored every minute using an Onset HOBO 64K Pendant® (Onset, Cape Cod, MA, USA) temperature data logger. Total ammonia nitrogen-N, nitrite-N, and nitrate-N were determined using a HACH DR/2400 spectrophotometer (HACH Company, Loveland, CO, USA). Alkalinity, as calcium carbonate, was measured using a HACH 16900 manual-digital titrator HACH Company, Loveland, CO, USA).

Fish were fed a standard pelleted tilapia maintenance diet containing approximately 35% protein and 6% fat (Zeigler Brothers, Inc., Gardners, PA, USA). Total fish acquired were 300 (84 × 3 groups + 18 controls = 270 + 30 extra = 300) where the additional 30 fish were included to account for any transportation mortality and/or unacceptable small-sized fish. All fish were handled and humanely euthanized using methods approved by Virginia Tech's Institutional Animal Care and Use Committee (VT-IACUC #19-155) under the auspices of Virginia Tech's Animal Welfare Assurance Program (#A-3208-01).

### 2.2. Treatment Groups

After a two-week acclimation period, fish were arbitrarily divided into three treatment groups consisting of 100 fish each and were placed into six separate 200–250 gallon tanks (50 fish/tank) labeled by treatment. Six fish from each treatment group were used as controls and were blood sampled and euthanized prior to treatment administration. Following treatment administration, 84 fish from each treatment group were sampled at various time points leaving approximately 10 fish remaining to account for unacceptable small-sized fish.

### 2.3. Dosing and Sampling

Flunixin meglumine (Banamine injectable solution; 50 mg/mL; Merck Animal Health, Madison, NJ, USA) was diluted with 0.9% NaCl to a ratio of 1:10 (1 part flunixin to 9 parts 0.9% NaCl) in a sterile container to reach a concentration of 5 mg/mL. The flunixin at a concentration of 5 mg/mL was then administered to fish by intramuscular injection at a dosage of 2.2 mg/kg.

Meloxicam (Metacam injectable; 5 mg/mL; Boehringer Ingelheim Vetmedica Inc., St. Joseph, MO, USA) was diluted with 0.9% NaCl to a ratio of 1:5 (1 part Meloxicam to 4 parts 0.9% NaCl) in a sterile container to reach a concentration of 1 mg/mL. The meloxicam at a concentration of 1 mg/mL was then administered to fish by intramuscular injection at a dosage of 1 mg/kg.

Meloxicam (Metacam oral; 1.5 mg/mL; Boehringer Ingelheim Vetmedica Inc.) was diluted with 0.9% NaCl to a ratio of 1:1.5 (1 part Meloxicam to 0.5 parts 0.9% NaCl) in a sterile container to reach a concentration of 1 mg/mL. The meloxicam at a concentration of 1 mg/mL was then administered to fish by oral gavage at a dosage of 1 mg/kg after a 24 h fasting period prior to dosing. Oral meloxicam was administered using a curved stainless steel 20-gauge 3" gavage tube (Popper and Sons, Inc., New Hyde Park, NY, USA) attached to a 100 μL Hamilton syringe. Gavage tube placement in the posterior portion of the stomach was confirmed manually.

For flunixin and meloxicam administration, fish were netted, individually weighed for dose determination, and manually restrained while being injected and gavaged, and immediately returned to the water. For each fish, this procedure took less than 15 s and no fish appeared distressed or suffered mortality as a result of this procedure. Prior to dosing, (time 0), six fish from each group were netted, sedated with sodium bicarbonate buffered (1:1) tricaine methanesulfonate (150 μg/L, MS-222, Sigma Scientific, St. Louis, MO) and bled. Blood was collected from the caudal tail vessels using a 23-gauge needle and syringe, and after sample collection fish were euthanized with buffered MS-222 (250 μg/L) followed by cervical separation to ensure death. Following flunixin or meloxicam administration, blood samples (~2 mL) were collected at 14 time points: 0.25, 0.5, 1, 2, 4, 6, 9, 12, 24 h, and 2, 4, 6, 8, and 10 days. At each sampling time, six fish from each group were netted, sedated with buffered MS-222, bled and then euthanized. Blood samples were immediately placed in individual plasma separator tubes containing lithium heparin (BD Microtainer, Becton, Dickinson and Company, Franklin Lakes, NJ, USA), mixed by inversion several times and kept on ice until centrifugation at $3000\times g$ for 10 min at 12 °C. Plasma samples were separated and placed in individual micro-centrifuge tubes and stored at −80 °C until analyzed for flunixin or meloxicam concentration. Following euthanasia, a sample of muscle tissue approximately 1.0–1.5 g was harvested from each fish. Skeletal muscle

ventral to the dorsal fin was harvested opposite the side of the intramuscular injection to avoid sampling the injection site. All samples were places in micro-centrifuge tubes and stored at −80 °C until analyzed for flunixin or meloxicam concentration.

*2.4. Flunixin and Meloxicam Determination*

2.4.1. Plasma Meloxicam and Flunixin Analysis

Plasma Extraction and Clean-Up

All solvents used such as methanol, acetonitrile and ammonium formate, formic acid were of LC–MS grade and purchased from Fisher Scientific (Hampton, NH, USA). Ultrapure 18 Ω water was obtained in house with a Millipore Synergy UV-R system. Concentrated phosphoric acid (~85%) was also purchase from Fisher Scientific. Meloxicam (MEL) and piroxicam (PIR, internal standard) were purchased from Cayman Chemicals (Ann Arbor, MI). 5′-Hydroxymethyl 5′-desmethyl meloxicam (HDM), a meloxicam metabolite, flunixin, 5-hydroflunixin and flunixin-d$_3$ standards were purchased from Toronto research Chemicals (Toronto, ON, Canada). Solid-phase extraction Elution plates Oasis PRIME HLB, 2 mg sorbent were purchased from Waters Co (Milford, MD, USA). Stock solutions of MEL and HDM at 100 g/mL were prepared in methanol and stored at −20 °C. A stock solution of the internal standard at 10 μg/mL was prepared in methanol and kept at −20 °C. For the calibration standards, working solutions of a mixture of HDM and MEL were prepared daily in aqueous ammonium formate 10 mM with 2% formic acid at the following concentrations: 1, 2.5, 5, 10, 25, 50, 100, 250, 500 and 1000 ng/mL. Quality controls plasma samples were prepared by spiking untreated fish plasma with HDM and MEL at the following concentrations: 4, 10 and 40 and 400 ng/mL and with 5-hydroxyflunixin and flunixin to at the following concentrations: 1.5, 15 and 150 ng/mL. A working solution of piroxicam at 100 ng/mL in aqueous ammonium formate 10 mM with 2% formic acid was prepared as well. All the samples were stored at −80 °C and thawed at room temperature (22 °C) for 20 min prior to extraction. The negative control plasma for meloxicam was prepared by adding 100 μL of untreated fish plasma, and 200 μL of aqueous ammonium formate 10 mM with 2% formic. The negative control plasma for flunixin was prepared by adding 100 μL of untreated fish plasma, and 200 μL of phosphoric acid 4%. For the calibration standards, 100 μL of untreated fish plasma was mixed with 100 μL of working standards and 100 μL of internal standard mixture at 100 ng/mL in at 100 ng/mL aqueous ammonium formate 10 mM with 2% formic for meloxicam or aqueous phosphoric acid 4% for flunixin. For the meloxicam samples and QCs, to 100 μL of fish plasma was added 100 μL of piroxicam working solution at 100 ng/mL and 100 μL of aqueous ammonium formate 10 mM with 2% formic and for the flunixin, 100 μL of fish plasma was added to 100 μL of flunixin-d$_3$ and 100 μL of aqueous phosphoric acid 4%. The samples were mixed and cleaned-up by solid-phase extraction. The negative control, standards, quality controls and samples were loaded on the SPE plate using positive pressure with nitrogen. Each well was washed twice with 0.25 mL of a mixture of methanol and water (25:75). The compounds were eluted with 2 × 25 μL aliquots of acetonitrile-methanol (90:10) and diluted with 50 μL of water with 0.1% formic acid before analysis.

Plasma Meloxicam LC–MS/MS Analysis

Samples were analyzed using a QExactive instrument (Thermo Fisher, Waltham, MA) connected to a Vanquish UPLC system. The chromatographic separation was performed using a column Agilent Technologies Eclipse Plus C18 2.1 × 100 mm, 1.8 μL with a gradient of aqueous formic acid 0.1% (A) and acetonitrile (B) as follows: from 0 to 0.5 min 30% B, at 4 min 80% B, from 4.010 to 4.50 min wash with 100% B followed by and a re-equilibration from 4.51 to 6.5 min with 30% B. The total run time was 6.5 min, the flow rate 0.4 mL/min, the column temperature was kept at 55 °C and the autosampler temperature at 10 °C. In these conditions, meloxicam, 5-hydroxymethyl-5′-desmethyl meloxicam and piroxicam retention times were 2.91 min, 2.17 min and 3.56 min, respectively. The analysis was performed in the positive mode using Parallel Reaction Monitoring (PRM). The system

was controlled through QExactive Tune 2.11 and TraceFinder software 4.1. The Heated Electrospray Ionization (HESI) ion source parameters were set as follows: sheath gas flow rate at 70, auxiliary gas flow rate at 20, sweep gas flow rate at 5, spray voltage at 2.5 kV, the capillary temperature at 280 °C and the auxiliary gas heater temperature at 400 °C. The PRM method combined two scan events starting with a full scan event followed by targeted MS/MS for the single charged precursor ions scheduled in an inclusion list. The full scan event employed a *m/z* 300–800 mass selection, an Orbitrap resolution of 35,000 at m/z 200, a target automatic gain control (AGC) value of $2 \times 10^5$, and maximum fill times (IT) of 50 ms. The targeted MS/MS was run at an Orbitrap resolution of 17,500 at *m/z* 200, a target AGC value of $1 \times 10^5$, and an IT time of 50 ms and an isolation window of 1.2 *m/z* unit window. MS/MS fragmentation was performed using the high-energy collision dissociation (HCD) mode, with normalized collision energy (NCE) of 50 eV for both meloxicam and its metabolite and 40 eV for the piroxicam (IS). Meloxicam, 5-hydroxymethyl-5′-desmethyl meloxicam and piroxicam were quantified using the products ions at *m/z* 115.032, 131.027 and 95.060, respectively. Identity of meloxicam and 5-hydroxymethyl-5′-desmethyl meloxicam were confirmed with the qualifier ions at *m/z* 141.012 and 157.007, respectively. The calibration curve was linear from 1 ng/mL to 1000 ng/mL for meloxicam and 5-hydroxymethyl-5′-desmethyl meloxicam with a minimum $R^2$ of 0.99 and a weighing factor of 1/x. Quality controls were used at 4 ng/mL, 40 ng/mL and 400 ng/mL with accuracies comprised between 80 and 120% and precision <15%.

Plasma Flunixin LC–MS/MS Analysis

The analysis was performed with a system from Waters Corporation (Milford, MA, USA) including an Acquity H UPLC and a TQ-S triple quadrupole mass spectrometer. Mass-Lynx and TargetLynx software 4.2 from Waters Co. (Milford, MA, USA), were used for the data acquisition and data analysis, respectively. The chromatographic separation was performed with a column Kinetex from Phenomenex (Torrance, CA, USA) 50 × 2.1 mm, 1.6 μL heated at 55 °C. The flow rate was set at 0.4 mL/min, the mobile phase consisted of a gradient of acetonitrile (B) and water containing 0.1% formic acid (A) as follows: 0 min: 30% B, 2.0 min: 70% B, 2.01 min: 100% B, 3.00 min: 30% B until 4 min. The total run time was 4 min. The injection volume was 5 μL and the temperature of the autosampler was maintained at 8 °C. The data acquisition was performed by Electrospray Ionization (ES) in the positive (ES+) mode using multiple reaction monitoring (MRM). The operating parameters for the mass spectrometer were as follows: the capillary voltage was set at 3.5 kV, the source and nitrogen desolvation temperatures were 150 °C and 600 °C, respectively. The desolvation nitrogen flow was set at 1000 L/h and the cone nitrogen flow at 150 L/h. The cone energy was set at 38 V for 5- hydroxyflunixin, 10 V for flunixin, and 42 V for flunixin-d$_3$. Helium was used as the collision gas. Desolvation and cone gas flow (nitrogen) were 1000 and 150 L/h, respectively and the collision gas flow (helium) was 0.15 mL/min. The data acquisition was performed by Electrospray Ionization in the positive mode (ESI+) using multiple reaction monitoring (MRM). From each precursor ion, two transition product ions were recorded, including one quantifier ion (Q) and one qualifier ion (q). The collision energy (CE) was expressed in volts (V). The dwell time was set automatically in order to get 20 points per peak. For flunixin, MRM transitions were *m/z* 297.144 > 279.1 (Q, CE = 20 V), *m/z* 297.1 > 259.1 (q; CE = 28 V). For 5-hydroxyflunixin, MRM transitions were *m/z* 313.1 > 295.1 (Q, CE = 22 v) and 313.1 > 197.0 (q; CE = 46 V). For flunixin-d$_3$, MRM transition was *m/z* 300.1 > 282.1 (Q, CE = 22 V). The calibration curve was linear from 1.0 ng/mL to 100 ng/mL for both flunixin and 5-hydroxyflunixin with a minimum $R^2$ of 0.99 and a weighing factor of 1/x. Quality controls were used at 1.5 ng/mL and 15 ng/mL, 150 ng/mL and 1000 ng/mL with accuracies comprised between 70 and 120% and precision <15%.

2.4.2. Tissue Meloxicam and Flunixin Analysis

Meloxicam Sample Extraction

Sample extraction for meloxicam followed the procedure described previously. Fish muscle tissue was weighted (1.0 g $\pm$ 0.1 g) in a 15 mL polypropylene tube and 50 µL of piroxicam (IS) 100 ng/mL in methanol was added. After mixing, 2 mL of 0.2 M sodium acetate buffer pH 4.75 was added. The mixture was homogenized and 25 µL of ß-glucuronidase (Sigma Aldrich, Catalog #G0876) was added and after mixing the tube was placed in an oscillating water bath at 37 °C for 1 h. After letting the tube cool down to room temperature, 5 mL of acetonitrile was added. The tube was shaken for 15 min and centrifuged at 2900× *g* for 7 min. The supernatant was transferred to a clean tube and the extraction repeated with 5 mL of acetonitrile, and the second extract being combined to the previous one. The final extract was centrifuged at 4500× *g* for 10 min before the clean-up steps. A 10 mL syringe was connected on top of an Alumina N column (Waters Co., P/N WAT020510). The extract was added to the syringe, the vacuum was turned on and the extract was passed through the Alumina column and collected in a clean 15 mL polypropylene tube. The acetonitrile was evaporated at 40 °C under vacuum concentrator until approximately a volume of 2 mL remained. The extract was centrifuged at 4500× g for 10 min and then cleaned-up on a Nexus C18 column 60 mg (Agilent Technologies, P/N 12103101). After conditioning the column with 2 mL of methanol followed by 2 mL of water, the aqueous extract was loaded on the column. The column was washed with 2 mL of water, let dry and then washed with 2 mL of *n*-hexane and let dry. Meloxicam and metabolite were eluted with 4 mL of hexane-ethyl acetate 50:50. The extract was evaporated at 40 °C under vacuum and reconstituted in 200 µL of acetonitrile-0.1% formic acid (30:70) following by centrifugation at 13,000× *g* for 10 min before being transferred to a HPLC vial.

Flunixin Sample Extraction

Sample extraction for flunixin and 5-hydroxyflunixin followed the procedure described below. All the solvents were of LC–MS grade. Flunixin, 5-hydroxyflunixin and flunixin-d3 were purchased from Toronto Research Chemicals (Toronto, ON, Canada). Working standards of flunixin and 5-hydroxyflunixin were prepared in a mixture of acetonitrile-water (4:1). Fish muscle tissues were weighted (1.0 g $\pm$ 0.1 g) in a 15 mL polypropylene tube. Additional drug free tilapia tissue from a commercially available source was homogenized and used as a negative control. For calibrators or QCs, 0.5 mL of the following working standards were added to 1.0 g of NEG CTRL homogenized tissue: 1, 5, 10, 25, 50, 100, 250, 500 and 1000 ng/mL. For quality controls (QCs), 0.5 mL of the following working standards were added: 2, 20, 200 and 800 ng/mL. then 0.5 mL of flunixin-d$_3$ at 100 ng/mL was added to each calibration, QC and sample. After mixing, 4 mL acetonitrile-water (4:1) were added and the tubes were shaken for 5 min followed by a centrifugation step at 4500× *g* for 5 min. The supernatant was transferred to a 15 mL polypropylene tube containing 0.25 g of C18 sorbent (Agilent, P/N5982-1188). Tubes were shaken for 30 s and centrifuged at 4500× *g* for 5 min and 0.1 mL of supernatant was transferred to a HPLC vial.

LC–MS/MS Analysis

Samples were analyzed using a QExactive instrument (Thermo Fisher) connected to a Vanquish UPLC system. The chromatographic separation was performed using a column Agilent Technologies Eclipse Plus C18 2.1 × 100 mm, 1.8 µL with a gradient of aqueous formic acid 0.1% (A) and acetonitrile (B) as follows: from 0 to 0.5 min 30% B, at 4 min 80% B, from 4.010 to 4.50 min wash with 100% B followed by and a re-equilibration from 4.51 to 6.5 min with 30% B. The total run time was 6.5 min, the flow rate 0.4 mL/min, the column temperature was kept at 55 °C and the autosampler temperature at 10 °C. In these conditions, meloxicam, 5-hydroxymethyl-5′-desmethyl meloxicam and piroxicam retention times were 2.91 min, 2.17 min and 3.56 min, respectively, and 5-hydroxyflunixin, flunixin and flunixin-d$_3$ retention times were 1.87, 2.02 min and 2.02 min, respectively.

The analysis was performed in the positive mode using Parallel Reaction Monitoring (PRM). The system was controlled through QExactive Tune and TraceFinder software. The Heated Electrospray Ionization (HESI) ion source parameters were set as follows for meloxicam: sheath gas flow rate at 70, auxiliary gas flow rate at 20, sweep gas flow rate at 5, spray voltage at 2.5 kV, the capillary temperature at 280 °C and the auxiliary gas heater temperature at 400 °C and as follows for flunixin: sheath gas flow rate at 70, auxiliary gas flow rate at 15, sweep gas flow rate at 5, spray voltage at 3.0 kV, the capillary temperature at 370 °C and the auxiliary gas heater temperature at 400 °C. The approximate room temperature of the laboratory was 21 °C. The PRM method combined two scan events starting with a full scan event followed by targeted MS/MS for the single charged precursor ions scheduled in an inclusion list.

For meloxicam, the full scan event employed a *m/z* 300–800 mass selection, an Orbitrap resolution of 35,000 at m/z 200, a target automatic gain control (AGC) value of $2 \times 10^5$, and maximum fill times (IT) of 50 ms. The targeted MS/MS was run at an Orbitrap resolution of 17,500 at *m/z* 200, a target AGC value of $1 \times 10^5$, and an IT time of 50 ms and an isolation window of 1.2 *m/z* unit window. MS/MS fragmentation was performed using the high-energy collision dissociation (HCD) mode, with normalized collision energy (NCE) of 50 eV for both meloxicam and its metabolite and 40 eV for the piroxicam (IS). Meloxicam, 5-hydroxymethyl-5′-desmethyl meloxicam and piroxicam were quantified using the products ions at *m/z* 115.032, 131.027 and 95.060, respectively. Identity of meloxicam and 5-hydroxymethyl-5′-desmethyl meloxicam were confirmed with the qualifier ions at *m/z* 141.012 and 157.007, respectively. The calibration curve was linear from 1.25 ng/mL to 50 ng/mL for meloxicam and from 5 to 50 ng/mL for 5-hydroxymethyl-5′-desmethyl meloxicam with a minimum $R^2$ of 0.99 and a weighing factor of $1/x$. Quality controls were used at 2 ng/g and 20 ng/g with accuracies comprised between 80 and 120% and precision <15%.

For flunixin, the full scan event employed a *m/z* 80–500 mass selection, an Orbitrap resolution of 17,500 at m/z 200, a target automatic gain control (AGC) value of $1 \times 10^6$, and maximum fill times (IT) of 50 ms. 5-Hydroxyflunixin, flunixin, and flunixin-$d_3$ precursor ions were *m/z* 313.079, 298.085, and 300.103, respectively. The targeted MS/MS was run with an Orbitrap resolution of 17,500 at *m/z* 200, a target AGC value of $1 \times 10^5$, and an IT time of 50 ms and an isolation window of 1.2 *m/z* unit window. MS/MS fragmentation was performed using the high-energy collision dissociation (HCD) mode, with normalized collision energy (NCE) of 40 eV for 5-hydroxyflunixin and a combination of 35 and 65 eV for flunixin. 5-Hydroxyflunixin, flunixin, and flunixin-$d_3$ were quantified using the products ions at *m/z* 280.045, 279.073 and 282.092, respectively. Identity of 5-hydroxyflunixin and flunixin were confirmed with the qualifier ions at *m/z* 295.0684 and 109.045, respectively. The calibration curve was linear from 2.5 ng/mL to 500 ng/mL for both flunixin and 5-hydroxyflunixin with a minimum $R^2$ of 0.99 and a weighing factor of $1/x$. Quality controls were used at 10 ng/g and 100 ng/g and 400 ng/g with accuracies comprised between 80 and 120% and precision <15%.

*2.5. Pharmacokinetic Analysis*

The plasma concentration versus time data were analyzed with a non-compartmental approach using a commercially available software (Phoenix®, Version 8.3, Certara, Inc., Princeton, NJ, USA). Mean concentration versus time profile for each treatment group was generated by taking the mean concentration value of the data from six fish for each time point. Only concentrations above the limit of quantification (LOQ) for all six fish each time point were included in the analysis. PK parameters were calculated based on the mean concentration versus time profile for each treatment group. The following PK parameters were calculated, including; slope of the terminal phase ($\lambda_z$), terminal half-life (T $\frac{1}{2}$), maximum plasma concentration ($C_{max}$); time to achieve peak concentration ($T_{max}$), time of last measurable (positive) concentrations in all six fish ($T_{last}$), area under the curve from the time of dosing ($Dosing_{time}$) to the last measurable (positive) concentration

($AUC_{0-last}$), AUC from $Dosing_{time}$ extrapolated to infinity, based on the last observed concentration (obs) ($AUC_{0-\infty}$), percentage of $AUCINF_{obs}$ due to extrapolation from $T_{last}$ to infinity ($AUC_{\%Extrap}$), volume of distribution based on the terminal phase per fraction of dose absorbed ($Vz\_F$), total body clearance per fraction of dose absorbed ($Cl\_F$), area under the moment curve from the time of dosing ($Dosing_{time}$) to the last measurable (positive) concentration ($AUMC_{0-last}$), area under the first moment curve (AUMC) extrapolated to infinity, based on the last observed concentration ($AUMC_{0-\infty}$), percent of $AUMC_{0-\infty}$ that is extrapolated ($AUMC_{\%Extrap}$), mean residence time from the time of dosing ($Dosing_{time}$) to the time of the last measurable concentration ($MRT_{last}$), and mean residence time (MRT) extrapolated to infinity ($MRT_{0-\infty}$). The $\lambda_z$ was calculated using linear regression of the terminal part of the log plasma concentration versus time curve and a linear trapezoidal linear interpolation method was used to determine $AUC_{0-last}$. The AUMC was calculated by combining the trapezoid calculation of $AUMC_{0-last}$ and extrapolated area. The MRT was calculated as MRT = AUMC/AUC and CL/F was calculated as CL = dose/AUC. The $C_{max}$ represented the observed peak plasma concentration, and the $T_{max}$ was the time to reach $C_{max}$.

## 3. Results

### 3.1. Fish and Environmental Parameters

Mean weights of the FLU, MEL-IM, and MEL-PO treatment groups at the time of drug administration were 297.0 ± 6.21 g, 296.0 ± 6.60 g, and 287.5 ± 6.80 g, respectively (Table 1). Overall mean ± SE of water quality parameters: temperature 25.5 ± 0.1 °C, dissolved oxygen 6.13 ± 0.07 mg/L, pH 7.95 ± 0.04, total ammonia-N 0.15 ± 0.03 mg/L, nitrite-N 0.16 ± 0.03 mg/L, nitrate-N 13.5 ± 5.4 mg/L, and alkalinity as CaCO3 79.3 ± 2.0 ppm.

**Table 1.** Experimental design and mean weights of the FLU, MEL-IM, and MEL-PO groups at the time of drug administration in tilapia.

| | Treatment Groups | | |
| --- | --- | --- | --- |
| | **FLU** | **MEL-IM** | **MEL-PO** |
| Number of tilapia per group (n) | 84 | 84 | 84 |
| Number of sample times collected | 14 | 14 | 14 |
| Number of fish sampled at each time point | 6 | 6 | 6 |
| Weight of tilapia (g) | Mean = 297.0 ± 6.21 Min = 194 Max = 440 | Mean = 296.0 ± 6.60 Min = 189 Max = 513 | Mean = 287.5 ± 6.80 Min = 185 Max = 467 |
| Dose (mg/kg) | 2.2 | 1.0 | 1.0 |
| Mortality during experiment | 0 | 0 | 0 |

### 3.2. Plasma Concentrations

The mean ± SEM plasma concentrations of drug at time intervals between 0.25 and 240 h are given in Table 2. The time points included are those at which at least 1 of the 6 fish sampled contained a drug concentration above the LOQ. The comparative log drug concentration versus time curves are shown in Figures 1 and 2. The mean profiles of the two meloxicam treatments showed a similar trend with oral administration reaching a higher peak concentration and both drug concentrations becoming non-detectable at 12 h (Figure 1). 5'Hydroxy-desmethyl-meloxicam was detectable out to 6 d post-drug administration. Flunixin drug concentrations reached a much higher peak concentration and indicated much slower excretion of the drug from the body, with drug concentrations detectable out to 96 h post-administration (Figure 2). 5-Hydroxyflunixin was detectable out to 24 h post-drug administration.

**Table 2.** Mean ± SEM parent drug and metabolite concentration in plasma after administration of a single dose of flunixin (2.2 mg/kg) intramuscularly or meloxicam (1 mg/kg) intramuscularly or orally (n = 6) in tilapia.

| Time (h) | Concentration (ng/mL) | | | | | | | | | | | | | | | | | |
|---|---|---|---|---|---|---|---|---|---|---|---|---|---|---|---|---|---|---|
| | FLU | | | FLU-OH | | | MEL-IM | | | MEL-IM-5HD | | | MEL-PO | | | MEL-PO-5HD | | |
| | n | Mean | SEM | n | Mean | SEM | n | Mean | SEM | n | Mean | SEM | n | Mean | SEM | n | Mean | SEM |
| 0.25 | 6 | 4375.79 | 171.40 | 6 | 30.71 | 5.53 | 3 | 2.87 | 0.26 | 6 | <LOQ | | 4 | 5.57 | 1.18 | 6 | <LOQ | |
| 0.5 | 6 | 4826.72 | 402.71 | 6 | 122.38 | 21.83 | 2 | 3.01 | 0.40 | 6 | <LOQ | | 5 | 16.49 | 6.38 | 3 | 42.71 | 16.03 |
| 1 | 6 | 4372.20 | 237.12 | 6 | 192.19 | 33.17 | 1 | 3.25 | | 6 | <LOQ | | 6 | 58.70 | 31.49 | 6 | 36.55 | 13.04 |
| 2 | 6 | 4303.00 | 172.82 | 6 | 219.42 | 55.10 | 6 | 11.29 | 2.30 | 3 | 4.45 | 1.17 | 5 | 72.22 | 15.40 | 5 | 32.71 | 11.62 |
| 4 | 6 | 2439.61 | 387.80 | 6 | 100.83 | 20.52 | 6 | 8.27 | 1.37 | 4 | 8.97 | 2.13 | 6 | 54.70 | 10.82 | 6 | 137.00 | 67.08 |
| 6 | 6 | 1368.40 | 245.71 | 6 | 51.66 | 9.61 | 6 | 8.26 | 1.62 | 3 | 23.61 | 10.89 | 6 | 34.13 | 5.72 | 6 | 66.25 | 27.56 |
| 9 | 6 | 439.50 | 110.87 | 6 | 20.54 | 5.10 | 6 | 5.68 | 1.07 | 3 | 5.16 | 1.82 | 6 | 9.19 | 2.51 | 5 | 26.58 | 8.57 |
| 12 | 6 | 243.49 | 67.55 | 6 | 4.53 | 0.92 | 6 | 4.88 | 0.94 | 4 | 9.65 | 4.63 | 3 | 3.91 | 0.51 | 4 | 11.77 | 3.37 |
| 24 | 6 | 55.95 | 12.26 | 4 | 2.60 | 0.50 | 6 | <LOQ | | 6 | 7.70 | 0.08 | 6 | <LOQ | | 6 | 9.58 | 1.22 |
| 48 | 6 | 7.61 | 1.52 | 6 | <LOQ | | 6 | <LOQ | | 5 | 7.61 | 0.01 | 6 | <LOQ | | 5 | 8.16 | 0.19 |
| 96 | 4 | 2.56 | 1.41 | 6 | <LOQ | | 6 | <LOQ | | 5 | 7.61 | 0.01 | 6 | <LOQ | | 5 | 7.94 | 0.20 |
| 144 | 6 | <LOQ | | 6 | <LOQ | | 6 | <LOQ | | 5 | 7.62 | 0.01 | 6 | <LOQ | | 5 | 7.72 | 0.05 |
| 192 | 6 | <LOQ | | 6 | <LOQ | | 6 | <LOQ | | | <LOQ | | 6 | <LOQ | | 6 | <LOQ | |
| 240 | 6 | <LOQ | | 6 | <LOQ | | 6 | <LOQ | | | <LOQ | | 6 | <LOQ | | 6 | <LOQ | |

Where n < 6, the remaining sample concentrations were < LOQ and the mean concentration is only based on sample concentrations >LOQ.

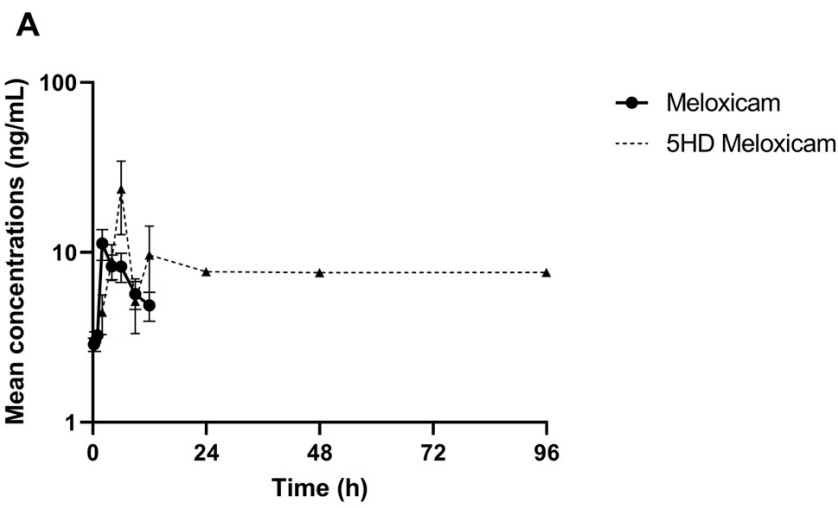

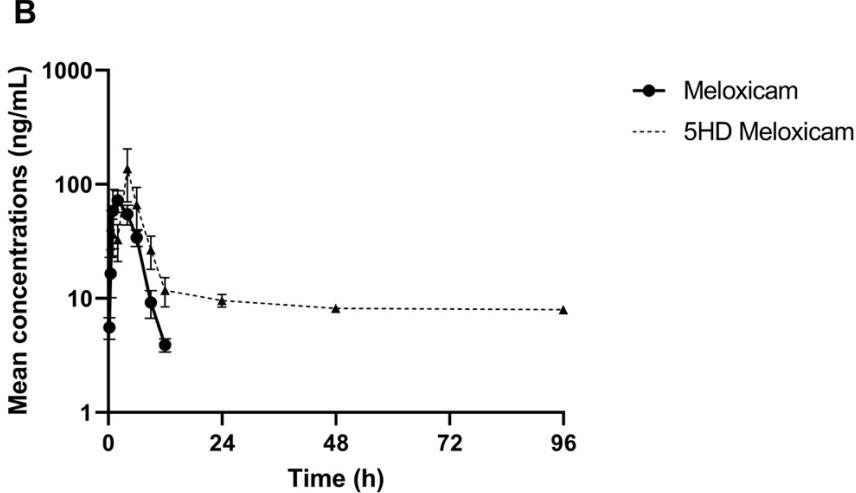

**Figure 1.** Mean ± SEM parent drug and metabolite concentration in plasma after administration of a single dose of meloxicam (1 mg/kg) orally (**A**) or intramuscularly (**B**) (n = 6) from 0 to 96 h post-administration in tilapia.

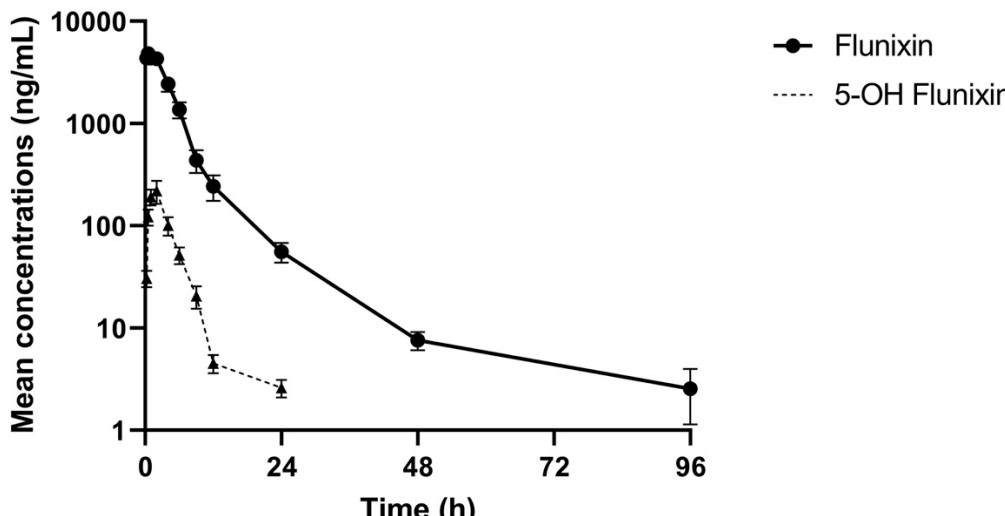

**Figure 2.** Mean ± SEM parent drug and metabolite concentration in plasma after administration of a single dose of flunixin (2.2 mg/kg) intramuscularly (n = 6) from 0 to 96 h post-administration in tilapia.

The PK parameters derived from plasma concentration versus time data of the three treatment groups are shown in Table 3. Drug absorption appeared to be quick for all three treatment groups with peak concentrations at 0.5 h for FLU and 2 h for both MEL-IM and MEL-PO. The $C_{max}$ of the three treatments were 4826.72 ng/mL, 11.29 ng/mL, and 72.22 ng/mL in the FLU, MEL-IM and MEL-PO groups, respectively and corresponding $T_{max}$ values were 0.5 h, 2 h, and 2 h. Plasma concentrations for the two meloxicam treatments were much lower than flunixin concentrations at all time points when concentrations were detectable. The $AUC_{0-\infty}$ (25261.62 h·ng/mL) from the FLU group was approximately 60 times higher than that obtained from the MEL-PO group (400.83 h·ng/mL) and 160 times higher than the $AUC_{0-\infty}$ obtained from the MEL-IM group (150.31 h·ng/mL). The $Cl_F$ of the FLU, MEL-IM, and MEL-PO groups were 87.09 mL/h/kg, 6653.13 mL/h/kg, and 2494.81 mL/h/kg, respectively, and the corresponding $T\frac{1}{2}$ were 7.34 h, 9.4 h, and 1.91 h.

**Table 3.** Pharmacokinetic parameters in plasma after administration of a single dose of flunixin (2.2 mg/kg) intramuscularly or meloxicam (1 mg/kg) intramuscularly or orally (n = 6) in tilapia.

| Parameter | Units | FLU | MEL-IM | MEL-PO |
|---|---|---|---|---|
| $\lambda_z$ | 1/h | 0.09 | NA | 0.36 |
| $T\frac{1}{2}$ | (h) | 7.34 | NA | 1.91 |
| $T_{max}$ | (h) | 0.5 | 2 | 2 |
| $C_{max}$ | (ng/mL) | 4826.72 | 11.29 | 72.22 |
| $T_{last}$ | (h) | 48 | 12 | 9 |
| $AUC_{0-last}$ | (h·ng/mL) | 25,180.93 | 84.13 | 375.54 |
| $AUC_{0-\infty}$ | (h·ng/mL) | 25,261.62 | NA | 400.83 |
| $AUC_{\%Extrap}$ | (%) | 0.32 | NA | 6.31 |
| $Vz_{\_F}$ | (mL/kg) | 922.84 | NA | 6868.6 |
| $Cl_{\_F}$ | (mL/h/kg) | 87.09 | NA | 2494.81 |
| $AUMC_{0-last}$ | (h·h·ng/mL) | 119,745.13 | NA | 1348.97 |
| $AUMC_{0-\infty}$ | (h·h·ng/mL) | 124,473.36 | NA | 1646.19 |
| $AUMC_{\%Extrap}$ | (%) | 3.8 | NA | 18.06 |
| $MRT_{last}$ | (h) | 4.76 | NA | 3.59 |
| $MRTI_{0-\infty}$ | (h) | 4.93 | NA | 4.11 |

NA indicates that this parameter value was not available because the calculated $AUC_{\%Extrap}$ was very high (i.e., >20%) and the terminal elimination phase following this administration regimen was not adequately characterized.

### 3.3. Tissue Concentrations

The mean ± SEM tissue concentrations of drug at time intervals between 0.25 h and 240 h are given in Table 4. The time points included are those at which at least 1 of the 6 fish sampled contained a drug concentration above the LOQ. Data were unable to be collected from the flunixin tissue samples for the 0.25 and 0.5 h time points.

The comparative tissue drug concentration versus time curves are shown in Figures 3 and 4 for each of the samples above the LOQ. The mean tissue profiles of the two meloxicam treatments show a similar trend to the plasma profiles with oral administration reaching a higher peak concentration and both drug concentrations becoming non-detectable at 9 h (Figure 3). 5'Hydroxy-desmethyl-meloxicam was not detected throughout this study following both oral and intramuscular administration. Flunixin tissue concentrations reached a higher peak concentration than meloxicam with tissue drug concentrations detectable out to 24 h post-administration (Figure 4). 5-Hydroxyflunixin was detectable out to 9 h post-drug administration.

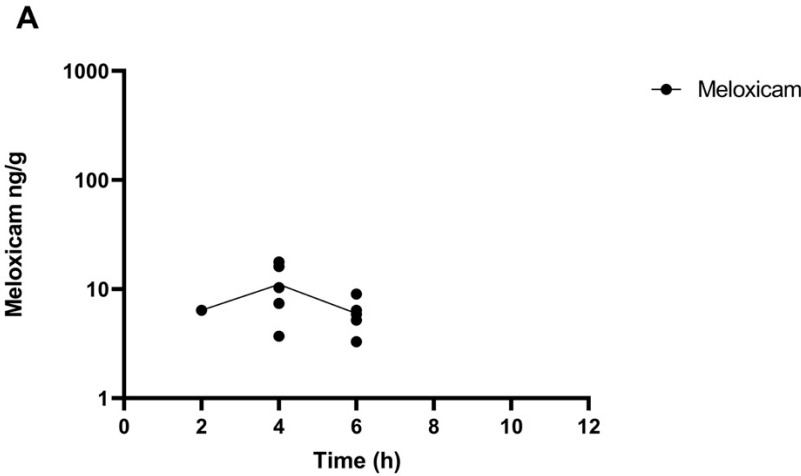

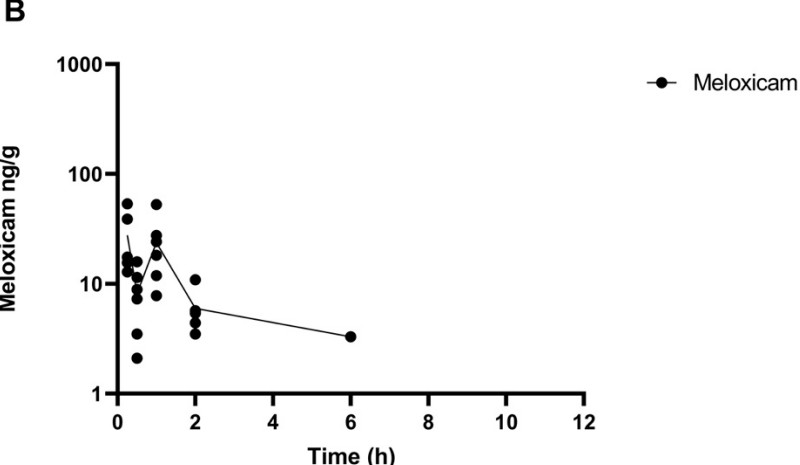

**Figure 3.** Mean ± SEM parent drug concentration in tissue after administration of a single dose of meloxicam (1 mg/kg) orally (**A**) or intramuscularly (**B**) (n = 6) from 0 to 12 h post-administration in tilapia.

**Table 4.** Mean ± SEM parent drug and metabolite concentration in tissue after administration of a single dose of flunixin (2.2 mg/kg) intramuscularly or meloxicam (1 mg/kg) intramuscularly or orally (n = 6) in tilapia.

| Time (h) | Concentration (ng/g) | | | | | | | | | | | | | | | | | |
|---|---|---|---|---|---|---|---|---|---|---|---|---|---|---|---|---|---|---|
| | FLU | | | FLU-OH | | | MEL-IM | | | MEL-IM-5HD | | | MEL-PO | | | MEL-PO-5HD | | |
| | n | Mean | SEM | n | Mean | SEM | n | Mean | SEM | n | Mean | SEM | n | Mean | SEM | n | Mean | SEM |
| 0.25 | | | | | | | 6 | <LOQ | | 6 | <LOQ | | 5 | 27.62 | 7.92 | 6 | <LOQ | |
| 0.5 | | | | | | | 6 | <LOQ | | 6 | <LOQ | | 6 | 8.18 | 2.08 | 6 | <LOQ | |
| 1 | 6 | 357.4 | 30.20 | 6 | 9.23 | 1.64 | 6 | <LOQ | | 6 | <LOQ | | 6 | 23.73 | 6.53 | 6 | <LOQ | |
| 2 | 6 | 461.65 | 46.18 | 6 | 19.5 | 5.14 | 1 | 6.4 | | 6 | <LOQ | | 4 | 6.125 | 1.64 | 6 | <LOQ | |
| 4 | 6 | 182.42 | 32.93 | 6 | 8.13 | 1.57 | 5 | 11.04 | 2.62 | 6 | <LOQ | | 6 | <LOQ | | 6 | <LOQ | |
| 6 | 6 | 93.37 | 22.44 | 5 | 5.98 | 0.78 | 5 | 5.96 | 0.93 | 6 | <LOQ | | 1 | 3.3 | | 6 | <LOQ | |
| 9 | 6 | 33.34 | 8.81 | 3 | 2.77 | 0.23 | 6 | <LOQ | | 6 | <LOQ | | 6 | <LOQ | | 6 | <LOQ | |
| 12 | 6 | 13.7 | 4.59 | 6 | <LOQ | | 6 | <LOQ | | 6 | <LOQ | | 6 | <LOQ | | 6 | <LOQ | |
| 24 | 3 | 2.07 | 0.74 | 6 | <LOQ | | 6 | <LOQ | | 6 | <LOQ | | 6 | <LOQ | | 6 | <LOQ | |
| 48 | 6 | <LOQ | | 6 | <LOQ | | 6 | <LOQ | | 6 | <LOQ | | 6 | <LOQ | | 6 | <LOQ | |
| 96 | 6 | <LOQ | | 6 | <LOQ | | 6 | <LOQ | | 6 | <LOQ | | 6 | <LOQ | | 6 | <LOQ | |
| 144 | 6 | <LOQ | | 6 | <LOQ | | 6 | <LOQ | | 6 | <LOQ | | 6 | <LOQ | | 6 | <LOQ | |
| 192 | 6 | <LOQ | | 6 | <LOQ | | 6 | <LOQ | | 6 | <LOQ | | 6 | <LOQ | | 6 | <LOQ | |
| 240 | 6 | <LOQ | | 6 | <LOQ | | 6 | <LOQ | | 6 | <LOQ | | 6 | <LOQ | | 6 | <LOQ | |

Where n < 6, the remaining sample concentrations were <LOQ and the mean concentration is only based on sample concentrations >LOQ.

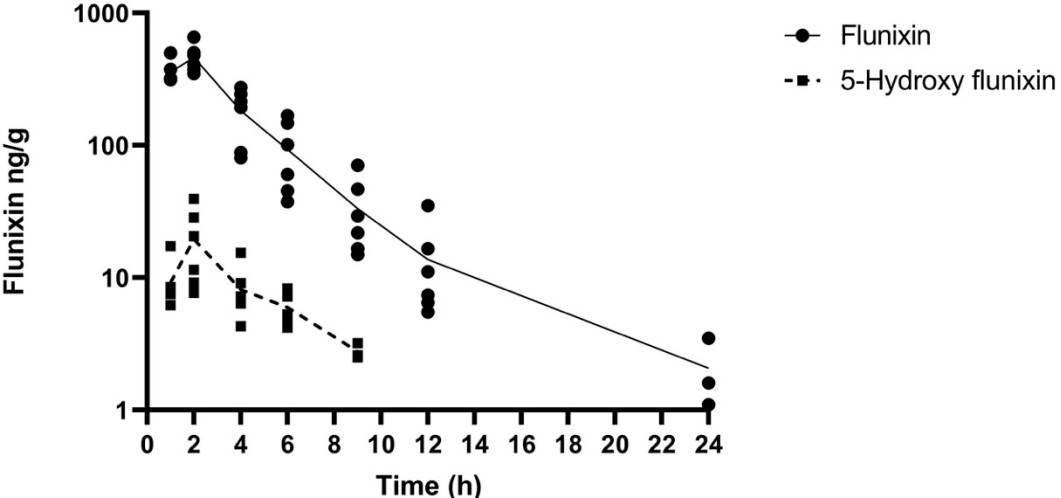

**Figure 4.** Mean ± SEM parent drug and metabolite concentration in tissue after administration of a single dose of flunixin (2.2 mg/kg) intramuscularly (n = 6) from 0 to 24 h post-administration in tilapia.

## 4. Discussion

There were no mortalities in any of the groups of fish, and no negative effects of drug treatment were observed during this study. Water quality variables during this study were within typical ranges for tilapia aquaculture [15]. A recent study found that MS-222 had no effect on florfenicol pharmacokinetics in Nile tilapia [16]. The use of MS-222 along with an NSAID is probable due to the widespread use of MS-222 as a sedative; there is no current data available on the effect of MS-222 on NSAID pharmacokinetics in Nile tilapia.

Tilapia belong to a class of fish known as cichlids from the family Cichlidae [17] which encompasses numerous species that are a part of aquaculture as well as the ornamental fish trade. Analgesics are an integral part of routine pain management in mammals, yet their use in aquaculture as well as the ornamental fish trade is still limited [8]. Research into post-operative analgesia investigating the use of an opioid or NSAID in ornamental fish (*Cyprinus carpio*) showed promise but has not been expanded to different species [18]. Fish that are subject to tissue damaging, invasive procedures, traumatic injury or aggression may require that pain and discomfort be reduced by the use of an analgesic such as an

NSAID; yet, validation of analgesic protocols is very limited and a great deal of species variation exists making extrapolation difficult [10]. Meloxicam has been administered intravenously and intramuscularly to Nile tilapia and reached relatively high plasma concentrations, but was eliminated faster than what is reported in the literature for other mammals [12]. Flunixin has been found to be an ineffective water treatment but has shown promise to reach a clinically effective concentration when administered intraperitoneally in catfish [11]. This study was the first to report on the pharmacokinetics of flunixin administered intramuscularly and meloxicam administered intramuscularly or orally at the given drug concentration and dosage in Nile tilapia.

Meloxicam, when administered intramuscularly or orally at a concentration of 1 mg/mL and a dosage of 1 mg/kg, likely did not reach clinically effective plasma concentrations and was quickly eliminated. When meloxicam was administered orally, the oral gavage tube was placed deep into the stomach and the volume of the drug was so small that no regurgitation occurred. The quick elimination is consistent with the findings of [12] when meloxicam was administered intravenously or intramuscularly at the same dose of 1 mg/kg, yet higher plasma concentrations were achieved compared to the present study. Meloxicam administered orally resulted in a larger AUC and longer terminal half-life than when administered intramuscularly, though both plasma concentrations were relatively low. 5'Hydroxy-desmethyl-meloxicam was quantified in the current study and was detectable in plasma out to 6 d post-drug administration for both intramuscular and oral routes indicative of a possible drug depot. The findings of the current study are consistent with previous findings suggesting that multiple daily administrations would be necessary to maintain a plasma concentration that could effectively control pain. Unfortunately, multiple dosing is impractical and stressful for large populations of fish maintained in aquaculture settings.

When flunixin was administered intramuscularly at a concentration of 50 mg/mL and a dosage of 2.2 mg/kg, relatively high plasma concentrations were achieved. Flunixin concentrations were detectable out to 96 h post-administration and 5-hydroxyflunixin was detectable out to 24 h. These findings are consistent with [11] who injected flunixin intra-peritoneally and reached a high plasma concentration deemed to be effective in catfish without signs of toxicity at a dosage of 2.2 mg/kg. Evaluation of clinical efficacy was outside the scope of this pharmacokinetic study, thus recommendations for effective dosing in Nile tilapia cannot be made. However, when administered intramuscularly, flunixin achieved a high $C_{max}$ (4826.72 ng/mL) and long enough terminal half-life (7.34 h) to potentially maintain clinically relevant plasma concentrations. Immediately following flunixin administration, an area of hyperpigmentation appeared around the injection site indicative of possible drug irritation which has also been reported in mammals upon intramuscular administration [19]. This localized darkening of the skin disappeared within 30 min from all fish involved in this study group.

Meloxicam concentrations were detectable in plasma for longer periods of time than in tissue for both oral and intramuscular administration. Meloxicam levels were below the LOQ of 2.5 ng/g at 8 h following drug administration and remained non-detectable. Flunixin peak concentrations in tissue were higher and detectable in tissue for a longer period of time than meloxicam tissue concentrations. Flunixin levels were below the LOQ of 2.5 ng/g at 48 h following drug administration and remained non-detectable. These results show a similar pattern of plasma concentrations for flunixin and meloxicam. However, these data suggest that meloxicam and flunixin reach non-detectable levels more quickly in tissue relative to plasma given the study conditions. The disparities between tissue and plasma data suggest that drug pharmacokinetics do differ between plasma and tissue. The meloxicam and flunixin metabolites quantified in this study (5'hydroxy-desmethyl-meloxicam and 5-hydroxyflunixin) have been shown to be active metabolites in cattle pharmacokinetic trials; however, [20,21] whether they are the most appropriate metabolites to quantify in fish has not been established. Further investigation into pharmacodynamics

and clinical efficacy via pain assessment and quantification of prostaglandin synthesis is warranted to build upon the results from the current study.

## 5. Conclusions

In conclusion, flunixin administered intramuscularly reached a sufficient plasma concentration to potentially have an analgesic effect, while meloxicam administered either intramuscularly or orally at the given dosage likely would not reach an analgesic concentration due to the relatively low plasma concentration. The feasibility of dosing individual fish is limited in commercial operations but may be relevant to settings where individual fish are more valuable and are handled on occasion. Development of an effective granular formulation of an NSAID would be more likely to be integrated into commercial operations. Further studies investigating different drug concentrations and dosage regimens of meloxicam, as well as clinical efficacy of flunixin and meloxicam in Nile tilapia are warranted to provide effective options for pain control in fish.

**Author Contributions:** Conceptualization, M.M., S.S. and J.C.; methodology, M.M., S.S., M.K. and J.C.; software, Z.L.; formal analysis, G.M., S.M. and Z.L.; investigation, M.M., S.S. and D.K.; resources, S.S., M.K. and J.C.; writing—original draft preparation, M.M., S.S., M.K. and J.C.; writing—review and editing, M.M., S.S., M.K., G.M., Z.L., D.K., S.M. and J.C.; supervision, S.S. and J.C.; funding acquisition, M.M. and J.C. All authors have read and agreed to the published version of the manuscript.

**Funding:** This project was supported by the College of Veterinary Medicine at Kansas State University. Miriam Martin is a Foundation for Food and Agriculture Research (FFAR) fellow supported by grant number 548795. Drs. Kleinhenz and Coetzee are supported by the Agriculture and Food Research Initiative Competitive, grant numbers 2017-67015-27124, 2020-67030-31479, 2020-67015-31540, 2020-67015-31546, and 2021-67015-34084 from the USDA National Institute of Food and Agriculture.

**Institutional Review Board Statement:** This study was conducted according to the guidelines of Virginia Tech's Institutional Animal Care and Use Committee (VT-IACUC #19-155).

**Acknowledgments:** The Phoenix® software license was provided to the Institute of Computational Comparative Medicine (ICCM) at Kansas State University by Certara USA, Inc. (Princeton, NJ, USA) as a part of the company's Academic Centers of Excellence program.

**Conflicts of Interest:** The authors declare no conflict of interest.

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
