# Peer review of "Comparative Pharmacokinetics and Tissue Concentrations of Flunixin Meglumine and Meloxicam in Tilapia (Oreochromis spp.)"

_fishes, doi:10.3390/fishes6040068_

Round 1
Reviewer 1 Report
It is a very interesting subject to study the application of analgesics in fish. The manuscript was to investigate the comparative pharmacokinetics of flunixin administered intramuscularly (IM) and meloxicam administered IM or orally (PO) in tilapia. However, in this paper, the author needs to explain the following problems.
- when oral administration of drugs, is there a "regurgitation" phenomenon? How to solve it?
- Why only the changes of drugs in plasma and muscle are measured? The data of drug concentration in liver, kidney and other organs can more completely reflect the metabolic process of drugs internal for pharmacokinetics.
- As the author discuss, the current lack of a validated approach to assessing pain in fish limits our ability to evaluate analgesic efficacy. Therefore, it may be more meaningful to study the remove of analgesics in fish (time to maintain a certain concentration).
Author Response
Line 484 of Discussion: Sentence added: When meloxicam was administered orally, the oral gavage tube was placed deep into the stomach and the volume of the drug was so small that no regurgitation occurred.
Reviewer 2 Report
General comments to fishes-1468017
The present work describes a comparative pharmacokinetics and tissue concentrations research of flunixin meglumine and meloxicam in tilapia fish. The manuscript is original, and might be useful to the scientific community. The manuscript topic is in line with the Fishes scope and deserves full consideration. The animal study design seems carried out correctly fully respecting the animal welfare. The animal sample size is very large (not negligible factor in such kind of research) and this produces pretty reliable findings.
The introduction is suitable for this topic and introduce well the use of pain killer drugs in fish. Material and method is very wide. All the needed information is given in detail and this allow readers to replicate the study. Results are very well presented and the discussion is conservative.
I just have a minor note. Table 3 reports the PK parameters of MEL-IM. In this analysis the value of AUC%extr is very high. All the PK parameters (excluded Tmax, Cmax and AUC0-last) make no sense and should be eliminated.
However, congrats to the authors for the exhaustive study.
With kind regards
Author Response
Table 3: All PK parameters excluding Tmax, Cmax, Tlast, and AUC0-last were replaced with NA and a footnote was added reading “NA indicates that this parameter value was not available because the calculated AUC%Extrap was very high (i.e., >20%) and the terminal elimination phase following this administration regimen was not adequately characterized”.